# Epidemiologic Implication of the Association between Herpes Simplex Virus Infection and the Risk of Type 1 Diabetes Mellitus: A Nationwide Case-Control Study in Taiwan

**DOI:** 10.3390/ijerph19137832

**Published:** 2022-06-26

**Authors:** Shao-Chang Wang, Jung-Yu Liao

**Affiliations:** 1Department of Laboratory Medicine, Kaohsiung Chang Gung Memorial Hospital, Kaohsiung 833401, Taiwan; adolphwang321@gmail.com; 2Department of Public Health, Kaohsiung Medical University, Kaohsiung 80708, Taiwan

**Keywords:** herpes simplex virus, human herpesvirus, type 1 diabetes mellitus, Taiwan’s National Health Insurance Research Database

## Abstract

Enterovirus infection is a known risk factor for type 1 diabetes (T1DM). Whether infection with other viruses induces T1DM remains undetermined. This study investigated the association between human herpesvirus (HHV) infection and the development of T1DM, using the data from Taiwan’s National Health Insurance Research Database. Patients with T1DM and age- and sex-matched controls were included. Subjects with HHV infection were subgrouped into those with histories of varicella-zoster virus, herpes simplex virus (HSV), Epstein-Barr virus, and human cytomegalovirus infections. The odds ratio of the risk of T1DM was calculated using a multivariable conditional logistic regression model. Atopic diseases, autoimmune thyroid diseases, and history of enterovirus infection served as adjusted comorbidities. Our findings suggested a significant association between HSV infection and the risk of T1DM (adjusted odds ratio: 1.21; 95% CI: 1.01–1.47, *p* = 0.048), while infection with other HHVs was not. The result of HSV infection remained significant when subjects were restricted to age ≤ 18 years (adjusted odds ratio: 1.35; 95% CI: 1.08–1.70, *p* = 0.010). We found a history of HSV infection might be an independent predictive risk factor for T1DM. This could be potentially helpful to the practice in public health.

## 1. Introduction

Type 1 diabetes mellitus (T1DM) is one of the most common autoimmune diseases in children and adolescents. The worldwide incidence of T1DM is approximately 15 per 100,000 people. The prevalence is estimated to be 9.5% in the world with variation in different countries [1]. There has been a global trend in the incidence increase in T1DM during the past three decades. A study investigated the trends in worldwide incidence of T1DM, including 57 countries, during 1990–1999, and found a significant increase in the incidence all over the world. The overall annual increase was 2.8%, and the increase during the period of 1995–1999 was even higher than the period of 1990–1994 (3.4% vs. 2.4%) [2]. Further studies revealed that the estimated annual incidence increased by 3.9% in Europe from 1989 to 2003 [3], and the annual increase in the U.S was 1.4%. from 2002 to 2012 [4]. This brought a clinical and public health burden owing to the long-term disease management with a huge demand for insulin. 

It is known that the mechanism of T1DM is the destruction of insulin-producing β-cells in the pancreatic islets caused by autoimmunity, which contributes to insufficient insulin secretion [5]. However, the development of T1DM is thought to be much more complex, with interactions between genetic and environmental factors playing important roles in the pathogenesis. Genetic variations in the HLA-DR and DQ genes are likely to contribute to the susceptibility or resistance to T1DM. Genotypes of HLA-DR1, DR3, DR4, and DR16 are considered to increase the susceptibility to T1DM, while HLA-DR11 and DR15 cause a resistant effect [6,7]. These genes encode HLA class II molecules which express on the surface of antigen-presenting cells, e.g., macrophages, B lymphocytes, and dendritic cells, and present antigens to CD4 T lymphocytes. The antigen-presenting interaction is considered the underlying process of all immune responses and may potentially cause an impact on the autoimmune response to the pancreatic β-cells [5].

The environmental factors that might potentially induce the development of T1DM include high birthweight, obesity, and cows’ milk, as well as deficiency of vitamin D and omega-3 fatty acids [8,9,10,11,12,13]. It has also been elucidated that virus-induced immune responses can lead to disorders in the immune system, which further participates in the complicated process of T1DM pathogenesis [14]. Among these, enterovirus infection has been widely explored in the literature for its association with T1DM [15]. In 1969, Gamble et al. first proposed the possibility of this association as they observed an increased incidence of T1DM following enterovirus infection with a seasonal variation pattern [16]. Further research based on both human and animal models, as well as prospective studies over the past decades, also showed similar results, especially the association with coxsackievirus B4 [17]. Viruses other than enteroviruses are also thought to be linked to T1DM. A few studies have suggested the potential roles of rotavirus, parvovirus, and encephalomyocarditis virus in the onset of T1DM [14]. Case reports have also demonstrated T1DM onset after mumps and H1N1 influenza infections [18,19]. Nevertheless, studies investigating the association between T1DM and viruses other than enteroviruses are limited.

Human herpesvirus (HHV) is a group of DNA viruses and includes varicella-zoster virus (VZV), herpes simplex virus (HSV), Epstein-Barr virus (EBV), and human cytomegalovirus (HCMV). HHV infections are widespread in humans. For example, research has shown more than 95% of adults of age 50 years or above were seropositive for VZV [20]; about 3.7 billion people were living with HSV type 1 in the world in 2016, while the number of people infected with HSV type 2 was estimated to be approximately 490 million [21]. Studies demonstrating the association between HHV and T1DM are scarce with limitations of small sample sizes or animal models [22,23,24]. This article aimed to investigate whether there is an association between HHV infection and the development of T1DM by analyzing a nationwide database, Taiwan’s National Health Insurance Research Database (NHIRD).

## 2. Materials and Methods

### 2.1. Subjects and Database Source

Data were obtained from the Taiwan Catastrophic Illness Patient Database (CIPD) and Longitudinal Health Insurance Database (LHID), which are both subsets of the NHIRD derived from Taiwan’s National Health Insurance (NHI) program. The NHIRD covers more than 99% of the medical record data of 23.38 million residents in Taiwan [25] and comprises robust medical information including patient demographics, dates of outpatient clinic visits and hospitalizations, diagnosis codes, administrated procedures, and prescribed medications. The information in the database was de-identified by labeling the patients in cryptographic code. The diagnosis codes were classified according to the International Classification of Diseases, Ninth Revision, Clinical Modification (ICD-9-CM). The CIPD contains patients diagnosed with specific catastrophic illnesses such as cancers and T1DM. The catastrophic illness certification exempts patients from National Health Insurance payments; thus, applications require conscientious expert review. Therefore, the diagnostic accuracy of T1DM is highly reliable in the CIPD. The LHID includes all NHIRD data from one million beneficiaries randomly selected from among the nationwide population between 1 January 1997, and 31 December 2013. The study was approved by the Kaohsiung Chang Gung Memorial Hospital institutional review board.

This section may be divided into subheadings. It should provide a concise and precise description of the experimental results, their interpretation, as well as the experimental conclusions that can be drawn.

### 2.2. Study Design

We conducted a case-control study. ICD-9-CM codes were used for the diagnosis of T1DM, history of HHV infection, and comorbidities. The details are listed in the Appendix A (Table A1, Table A2 and Table A3). Cases with T1DM from 1 January 2001, to 31 December 2013, were identified from the CIPD according to the ICD-9-CM codes. Age was recorded as of the date of diagnosis of T1DM, which was defined as the index date. Beneficiaries in the LHID served as candidates for control subjects after the exclusion of beneficiaries in the database with a diagnosis of T1DM. Each case with T1DM was matched to four control subjects with the same birth year and sex. A history of HHV infection was defined as the presence of outpatient HHV-related ICD-9-CM codes at least twice within 180 consecutive days or an inpatient HHV-related ICD-9-CM code once before the index day. HHV infection was classified into five subgroups according to the disease entities with related viruses and the ICD-9-CM codes: (1) VZV, varicella, (2) VZV, zoster, (3) HSV, (4) EBV, and (5) HCMV. Three atopic diseases, including allergic rhinitis, asthma, and atopic dermatitis, two autoimmune thyroid diseases, including Graves’ disease and Hashimoto’s thyroiditis, as well as history of enterovirus infection were considered as potential comorbidities [15,17,26,27,28], based on the related ICD-9-CM codes.

### 2.3. Statistical Analysis

The distributions of the defined variables were described as case numbers and proportions (percentages). The subjects were grouped according to age as follows: <1 year, 1–3 years, 3–5 years, 5–10 years, 10–18 years, and >18 years. Chi-square tests were used to compare the differences in the variables between the T1DM and non-T1DM groups. We estimated the risk of developing T1DM by performing multivariable conditional logistic regression with adjustment for the comorbidities of atopic diseases, autoimmune thyroid diseases, and history of enterovirus infection. Cases without a history of HHV infection and comorbidities were defined as the reference groups for each corresponding variable. The same multivariable conditional logistic regression model was repeated based only on subjects ≤18 years of age. The results were expressed as crude and adjusted odds ratios with respective 95% confidence intervals. All statistical analyses were conducted using SAS version 9.4; (SAS Institute, Inc., Cary, NC, USA). Statistical significance was defined as a two-sided *p*-value of <0.05.

## 3. Results

The flow chart of the study subject inclusion and exclusion is presented in Figure 1. We finally enrolled 8179 patients with T1DM and 32,716 age- and sex-matched controls. Women represented 52.7% of all subjects (Table 1). Regarding the distribution of age at T1DM diagnosis, 58.6% of the subjects were >18 years of age, 24.5% were 10–18 years of age, and 12.1% were 5–10 years of age. Fewer than 5% of subjects were diagnosed with T1DM when they were ≤5 years of age. Although the prevalence of HHV infection in the T1DM group was higher than that in the control group, the difference was not statistically significant (6.1% vs. 5.6%, *p* = 0.074). Regarding the subgrouping of HHV infection, the “VZV, varicella” subgroup accounted for the highest proportion of subjects (2.90% in the cases and 2.96% in the controls), followed by HSV (1.82% in the cases and 1.46% in the controls) and “VZV, zoster” (1.34% in the cases and 1.08% in the controls). Less than 0.1% of the subjects had a history of EBV or HCMV infections. Regarding the comorbidities of atopic diseases, both asthma and atopic dermatitis were significantly more common in the cases than in the controls. However, this difference was not significant for allergic rhinitis. The prevalence of autoimmune thyroiditis, Graves’ disease, and Hashimoto’s thyroiditis was significantly higher in patients with T1DM than that in controls without T1DM (4.2% vs. 0.3%, *p* < 0.001 and 1.6% vs. 0.1%, *p* < 0.001, respectively). The prevalence of a history of enterovirus infection was also significantly higher in the group of cases with T1DM than in the control group (7.3% vs. 6.5%, *p* = 0.008).

The results of multivariable conditional logistic regression showed no significantly elevated risk of T1DM in the patients with VZV, EBV, and HCMV infection, compared to subjects without a history of HHV infection. However, the patients with a history of HSV infection had a significantly higher risk of T1DM than did non-HHV subjects, with an odds ratio of 1.21 (95% CI: 1.01–1.47, *p* = 0.048) after adjusting for the comorbidities (Table 2). The statistical significance remains when estimating the risk based on the subjects ≤18 years of age (adjusted odds ratio: 1.35; 95% CI: 1.08–1.70, *p* = 0.010) (Table 3). The comorbidities associated with a significantly increased risk of T1DM were asthma, atopic dermatitis, Graves’ disease, Hashimoto’s thyroiditis, and history of enterovirus infection, while allergic rhinitis was not correlated with the outcome of T1DM.

## 4. Discussion

In the present study, the association between HHV infection and the onset of T1DM was evaluated using a nationwide, population-based dataset, the NHIRD. The results demonstrated a significant association between a history of HSV infection and the development of T1DM (adjusted odds ratio: 1.21; 95% CI: 1.01–1.47) compared to people without a history of HHV infection. The phenomenon is even more significant for people of age 18 years or less (adjusted odds ratio: 1.35; 95% CI: 1.08–1.70). Therefore, HSV infection might be an independent predictive factor for the development of T1DM. Infection by other HHVs, including VZV, EBV, and HCMV, on the other hand, was not obviously associated with the risk of T1DM. To our knowledge, this is the first study to use a nationwide database to explore the relationship between HHV infection and the risk of T1DM. Taiwan’s NHIRD covers more than 99% of the residents in Taiwan [25]. The nearly universal coverage minimized the self-selection bias. The diagnosis of T1DM was likely to be highly accurate because of the strict review of the registration for catastrophic illness certification in Taiwan’s National Insurance program.

The incidence of T1DM increased in children and young adults worldwide in the past three decades [2,3,4]. The mechanisms underlying the development of T1DM are complex, with genetic and environmental factors playing potential roles in its pathogenesis. Viral infection has been proposed as a risk factor for the development of T1DM. Among these, enterovirus infection is the most well-demonstrated [15,17]. Past cross-sectional studies have shown that the antibodies against enteroviruses were significantly more frequent in the patients with recently onset T1DM than the non-T1DM controls [29,30,31]. Prospective studies with a follow-up of a series of serum samples and more rigorous diagnostic criteria for enterovirus infection revealed a temporal correlation between the onset of T1DM and preceding evidence of enterovirus infection within 6 months, while the controls without evidence of T1DM had a significantly lower prevalence of experiencing enterovirus infection [32,33]. Investigations based on molecular tests also disclosed the association between the onset of T1DM and the presence of enterovirus RNA in the serum [34,35]. In addition, a study with the mouse model indicated that an infection of enterovirus could lead to β-cell damage as well as hypoinsulinemia and hyperglycemia [36,37]. Among these, Coxsackievirus B was the most widely studied member of enteroviruses, especially the Coxsackievirus B4 [29,36,37].

Few studies have reported on the association between T1DM and viruses other than enteroviruses. Regarding HHV, an early study found the frequency of the HCMV genome detected by molecular hybridizations in patients with newly diagnosed T1DM was much higher than the healthy control subjects (22% vs. 2.6%). Also, CMV genome-positive patients have higher islet cell antibodies and β-cell surface antibodies in their serum compared to the control group (62% vs. 33% and 69% vs. 33%, respectively) [22]. These findings suggested a strong correlation between HCMV infection and T1DM. However, the study included a relatively small number of cases (59 patients with T1DM and 38 control subjects). An experimental study in 2003 suggested rat cytomegalovirus infection could modulate the immunity of the rats to an accelerated diabetes development by the proliferation of T-cells [23]. The study was primarily based on an animal model, and part of these tests was performed in vitro. Whether HCMV induces an immune response to accelerate the development of T1DM in humans is still pending. When it comes to EBV, a study investigated antiviral antibodies against 646 viral antigens in patients with newly onset T1DM. The corresponding prevalence of the antiviral antibodies based on epidemiological studies was also taken into consideration. They found the antibodies against EBV were significantly higher in the case group than in the age- and sex-matched healthy controls (odds ratio: 6.6; 95% CI 2.0–25.7) [24]. Nevertheless, the sample size of the study was small (42 cases and 42 controls). There were also case reports that suggested the possible association between HHV infection and the development of T1DM. Fujiya et al. reported that a 70-year-old woman with multiple myeloma experienced an onset of fulminant T1DM followed by reactivation of EBV after chemotherapy [38]. All these studies have some limitations, including small sample sizes, restricted case reports, and animal models. An investigation based on a large human sample size is needed to further determine the association between HHV infection and the risk of T1DM.

Studies have focused on the mechanism that induces the development of T1DM by viruses. Several studies proposed a theory of “cross-reactivity by molecular mimicry” based on the similarities in molecular structures between virus antigens and human autoantigens. Parkkonen et al. showed that an amino acid sequence (GPPAA) of the HLA-DQ8 β chain, which is important in defining the risk for T1DM, also exists in the EBNA3C protein of EBV in six successive repeats [39]. Both human sera and affinity-purified antibodies against this EBV-derived peptide react with the peptide in the HLA-DQ8 β chain. Hiemstra et al. also reported in vitro T-cell cross-reactivity between the HCMV major DNA-binding protein and glutamic acid decarboxylase 65 (GAD65)-specific T cells [40]. GAD65 is a neuroenzyme expressed in pancreatic β-cells and is thought to be an autoantigen of T1DM. Similar molecular mimicries were also observed in the sequence similarities between a GAD65 epitope and the 2C protein coxsackievirus [14], as well as the pancreatic islet tyrosine phosphatase islet antigen 2 (IA-2) molecule and the rotavirus VP7 protein [41]. Another theory, the “bystander activation of self-reactive nonspecific T cells”, suggests that pancreatic islet cells exhibit cytolysis during enterovirus infection. Thus, the hidden self-components are exposed. These endogenous antigens might be recognized by antigen-presenting cells and are then presented to self-reactive T cells that do not belong to specific T-cell receptor stimulation. The activated naive islet-specific T cells may further contribute to islet β-cell destruction and the development of T1DM [15].

HSV is a double-stranded DNA virus and is classified into HSV types 1 and 2. These types share 50% sequence homology and cross-react serologically. However, each has a unique protein. HSV grows rapidly and causes cytolysis in the host. These viruses are widespread in humans and are responsible for a variety of clinical entities via primary infection or the reactivation of latent infection [42]. Symptoms of HSV type 1 infection are typically gingivostomatitis and vesicular or ulcerative lesions over the oropharyngeal mucosa, while HSV type 2 infection causes genital herpes. Other clinical manifestations include keratoconjunctivitis and cutaneous lesions. Meningitis and encephalitis could also occur. Understanding the human immune system activation by these HSV-induced diseases might be helpful to further elucidate their possible association with the pathogenesis of T1DM.

Our results suggested that the incidence of comorbidities such as asthma, atopic dermatitis, Graves’ disease, Hashimoto’s thyroiditis, and history of enterovirus infection were higher in patients with T1DM compared to those without T1DM, while allergic rhinitis was not. These findings were consistent with the previous studies reporting a heterogeneous association between T1DM and atopic diseases after meta-analysis and a strong positive association between T1DM and autoimmune thyroid diseases [26,27,28].

Our study has several limitations. First, we did not include sociodemographic data, such as educational or socioeconomic status. Additionally, there was no information on family history, birthweight, physical activities, exercise habits, or body mass index (BMI), as well as dietary factors like milk and cod liver oil. These factors could be potentially associated with the outcome but we were unable to obtain these data for our analyses because of no information in the NHIRD. Second, we could not accurately record the time of the onset of T1DM. Alternately, we analyzed the date when T1DM was diagnosed. Generally, the time interval between the disease onset and diagnosis may not be long, the date we analyzed did not perfectly reflect the true time of the disease onset. Third, the diagnosis of HHV infection is based on the diagnosis code of ICD-9-CM. No laboratory data were available to support the diagnosis owing to the absence of such records in the database; however, the data provided from the NHIRD are highly accurate, which has been proved by previous studies [43,44]. The Bureau of NHI endeavored the quality of NHIRD [45]. In addition, many people with HHV infection are asymptomatic, especially those with latent infections, which may contribute to an underestimation of the number of cases of HHV infection. However, our study included patients with clinical symptoms whose active disease status was more likely to trigger an immune response and, thus, was more potentially associated with the development of T1DM than people with latent infection. Fourth, the time elapsed from virus infection to further immune disorders and the development of T1DM is currently not fully understood. It may range from short to quite long, a few days to several months or even years for example. The database we used comprised a long period of more than 15 years (from 1 January 1997 to 31 December 2013) which could attenuate the effect of this limitation. Fifth, the results of our study did not prove a causal relationship between HSV infection and the pathogenesis of T1DM. While it is hypothesized that viral infection leads to immune disorders and T1DM, convincing scientific evidence is required.

## 5. Conclusions

The results of our study suggested that HSV infection is associated with the risk of the development of T1DM. This finding might be potentially helpful to the application in disease prevention and health promotion. Studies focusing on this topic in the literature are very limited. Population-based investigations in different countries are warranted. Moreover, further scientific evidence of the relationship between HSV infection and T1DM should be explored to improve our understanding of the pathogenesis of T1DM.

## Figures and Tables

**Figure 1 ijerph-19-07832-f001:**
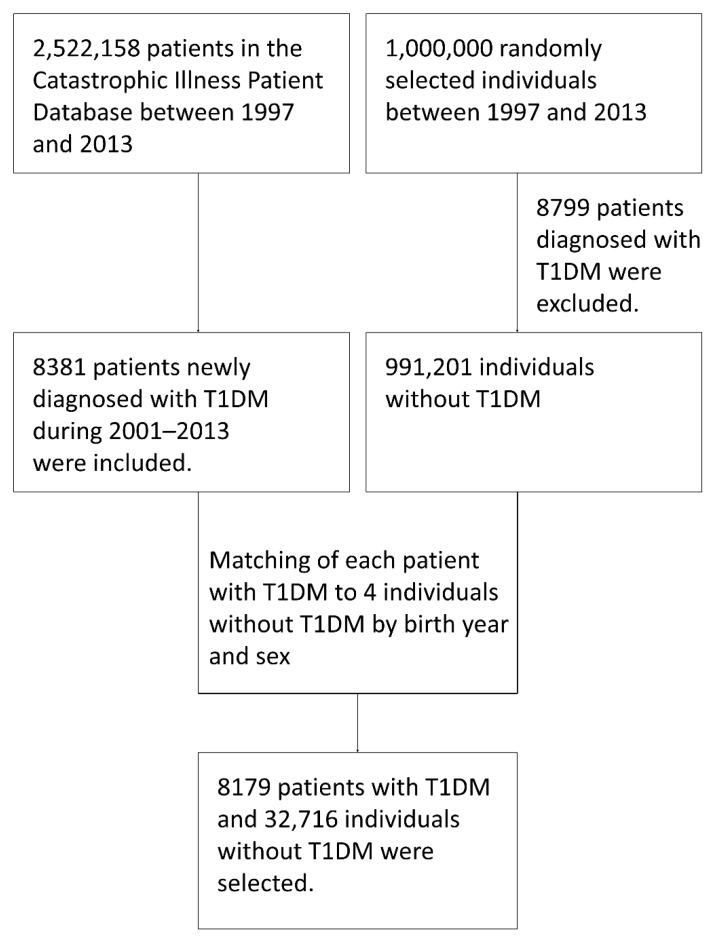
Flowchart of subject inclusion and exclusion. T1DM, type 1 diabetes mellitus.

**Table 1 ijerph-19-07832-t001:** Distribution of selected characteristics of 8179 cases and 32,716 controls.

	T1DM		Non-T1DM		
Variable	*n*	(%)	*n*	(%)	*p*-Value
Total	8179		32,716		
Gender					1.0
Female	4309	(52.7)	17,236	(52.7)	
Male	3870	(47.3)	15,480	(47.3)	
Age					1.0
≤1	25	(0.3)	100	(0.3)	
1–3	149	(1.8)	596	(1.8)	
3–5	217	(2.6)	868	(2.6)	
5–10	989	(12.1)	3956	(12.1)	
10–18	2002	(24.5)	8008	(24.5)	
>18	4797	(58.6)	19,188	(58.6)	
History of HHV infection					0.074
VZV, varicella	237	(2.90)	968	(2.96)	
VZV, zoster	110	(1.34)	352	(1.08)	
HSV	149	(1.82)	479	(1.46)	
EBV	6	(0.07)	26	(0.08)	
HCMV	1	(0.01)	3	(0.01)	
No	7676	(93.9)	30,888	(94.4)	
Allergic rhinitis					0.474
Yes	1523	(18.6)	5980	(18.3)	
No	6656	(81.4)	26,736	(81.7)	
Asthma					0.005 *
Yes	772	(9.4)	2768	(8.5)	
No	7407	(90.6)	29,948	(91.5)	
Atopic dermatitis					<0.001 *
Yes	296	(3.6)	939	(2.9)	
No	7883	(96.4)	31,777	(97.1)	
Graves’ disease					<0.001 *
Yes	341	(4.2)	111	(0.3)	
No	7838	(98.0)	32,605	(99.7)	
Hashimoto’s thyroiditis					<0.001 *
Yes	134	(1.6)	35	(0.1)	
No	8045	(98.4)	32,681	(99.9)	
History of enterovirus infection					0.008 *
Yes	595	(7.3)	2115	(6.5)	
No	7584	(92.7)	30,601	(93.5)	

Abbreviations: EBV, Epstein-Barr virus; HCMV, human cytomegalovirus; HHV, human herpesvirus; HSV, herpes simplex virus; T1DM, type 1 diabetes mellitus; VZV, varicella zoster virus. Statistical significance was set at *p* < 0.05. * indicates statistical significance.

**Table 2 ijerph-19-07832-t002:** Estimating risk of T1DM by multivariable conditional logistic regression.

	Crude		Adjusted	
Variable	Odds Ratio (95% CI)	*p*-Value	Odds Ratio (95% CI)	*p*-Value
HHV type				
VZV, varicella	0.99 (0.85–1.14)	0.865	0.97 (0.84–1.13)	0.691
VZV, zoster	1.26 (1.01–1.57)	0.037 *	1.17 (0.94–1.47)	0.167
HSV	1.26 (1.04–1.51)	0.018 *	1.21 (1.01–1.47)	0.048 *
EBV and HCMV	0.97 (0.43–2.21)	0.942	0.95 (0.42–2.18)	0.911
No	Reference		Reference	
Allergic rhinitis				
Yes	1.03 (0.96–1.10)	0.450	0.97 (0.91–1.05)	0.459
No	Reference		Reference	
Asthma				
Yes	1.14 (1.05–1.25)	0.003 *	1.13 (1.02–1.24)	0.014 *
No	Reference		Reference	
Atopic dermatitis				
Yes	1.28 (1.12–1.46)	<0.001 *	1.23 (1.07–1.41)	0.004 *
No	Reference		Reference	
Graves’ disease				
Yes	14.5 (11.5–18.3)	<0.001 *	13.7 (10.8–17.3)	<0.001 *
No	Reference		Reference	
Hashimoto’s thyroiditis				
Yes	15.3 (10.6–22.2)	<0.001 *	13.1 (8.9–19.2)	<0.001 *
No	Reference		Reference	
History of enterovirus infection				
Yes	1.19 (1.06–1.32)	0.002 *	1.16 (1.04–1.30)	0.008 *
No	Reference		Reference	

Abbreviations: EBV, Epstein-Barr virus; HCMV, human cytomegalovirus; HHV, human herpesvirus; HSV, herpes simplex virus; VZV, varicella zoster virus. Statistical significance was set at *p* < 0.05. * indicates statistical significance.

**Table 3 ijerph-19-07832-t003:** Estimating risk of T1DM by multivariable conditional logistic regression model in subjects of age ≤ 18.

	Crude		Adjusted	
Variable	Odds Ratio (95% CI)	*p*-Value	Odds Ratio (95% CI)	*p*-Value
HHV type				
VZV, varicella	1.01 (0.86–1.18)	0.946	0.98 (0.84–1.16)	0.813
VZV, zoster	1.19 (0.68–2.09)	0.539	1.18 (0.67–2.08)	0.579
HSV	1.39 (1.11–1.75)	0.004 *	1.35 (1.08–1.70)	0.010 *
EBV and HCMV	0.60 (0.21–1.71)	0.337	0.59 (0.21–1.68)	0.320
No	Reference		Reference	
Allergic rhinitis				
Yes	1.08 (0.99–1.18)	0.094	1.04 (0.94–1.14)	0.576
No	Reference		Reference	
Asthma				
Yes	1.09 (0.98–1.21)	0.127	1.03 (0.92–1.16)	0.466
No	Reference		Reference	
Atopic dermatitis				
Yes	1.23 (1.04–1.46)	0.017 *	1.20 (1.01–1.42)	0.042 *
No	Reference		Reference	
Graves’ disease				
Yes	20.7 (8.6–49.5)	<0.001 *	18.1 (7.5–43.8)	<0.001 *
No	Reference		Reference	
Hashimoto’s thyroiditis				
Yes	27.3 (11.6–64.4)	<0.001 *	24.9 (10.5–58.9)	<0.001 *
No	Reference		Reference	
History of enterovirus infection				
Yes	1.19 (1.07–1.34)	0.002 *	1.17 (1.04–1.31)	0.007 *
No	Reference		Reference	

Abbreviations: EBV, Epstein-Barr virus; HCMV, human cytomegalovirus; HHV, human herpesvirus; HSV, herpes simplex virus; VZV, varicella zoster virus. Statistical significance was set at *p* < 0.05. * indicates statistical significance.

## Data Availability

Data are available from the National Health Insurance Research Database (NHIRD) published by the Taiwan National Health Insurance (NHI) Administration. Due to legal restrictions imposed by the government of Taiwan concerning the “Personal Information Protection Act”, data cannot be made publicly available.

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
