# Peer review of "Epidemiologic Implication of the Association between Herpes Simplex Virus Infection and the Risk of Type 1 Diabetes Mellitus: A Nationwide Case-Control Study in Taiwan"

_ijerph, 2022, doi:10.3390/ijerph19137832_

Round 1

Reviewer 1 Report

The manuscript entitled “Association between herpes simplex virus infection and the risk of type 1 diabetes mellitus: a nationwide case-control study in Taiwanwith the aim to investigate the association 11 between human herpesvirus (HHV) infection and the development of T1DM, using the data from Taiwan’s National Health Insurance Research Database.

Major Critique:  

1.     Quoted in the manuscript of Introduction “In the literature, enterovirus infection has been widely explored as its association with T1DM, and research based on both human and animal models as well as prospective studies over the past decades also showed similar results, especially the association with coxsackievirus B4.” Also ‘history of enterovirus infection’ is remained as a significant association in your analysis, however, is there any reason why the authors did not mention a word in the Abstract or in the Discussion? Please revise the manuscript focus on related references and possible factors should be added. Please elucidate the confounding factor of enterovirus infection?

2.     I'm wondering if the authors can explore the potential interaction effects between the history of HSV infection and history of enterovirus.

3.  The diagnostic accuracy of T1DM is highly reliable in the Taiwan Catastrophic Illness Patient Database. Is also there a highly reliable diagnosis or information about ‘The history of HHV infection’ in your study? Is there any references to support study design?  

Reviewer 2 Report

The authors present a retrospective analysis of the association of HHV infections with T1DM. A few points to consider:

-minor: an article instruction heading was left in on the top of page 3, please remove.

-for the matching, did you ensure a match was not positive for HHV ICD-9 codes as well? Should matching also include weight or exclude factors for matching health participants that could bias the matching process?

-Is family history available in this database to rule out the effect of a family history of T1DM in cases/controls?

-would ICD-9-CM codes always capture HHV or should you consider including medications used for these infections in your identification of associations? This may be an important consideration as there is some data suggesting that some of these medications are associated with insulin resistance.

-in the table footnotes for your regressions, suggest adding in the variables you are controlling for.

-was there any correlation in the data between HHV infection, T1DM and age of onset?

Round 2

Reviewer 1 Report

The manuscript entitled Association between herpes simplex virus infection and the risk of type 1 diabetes mellitus: a nationwide case-control study in Taiwanwith the aim to investigate the association between human herpesvirus (HHV) infection and the development of T1DM, using the data from Taiwan’s National Health Insurance Research Database.

1.     The basic question is interesting and the data are consistent with most of the notion following revision. However, there are still some minor points need to be addressed.

2.     Headline: " Epidemiologic implication of the association between herpes simplex virus infection and the risk of type 1 diabetes mellitus: a nationwide case-control study in Taiwan " - might be a better title in my personal opinion.

3.     Is there any reference about the criteria to support HHV diagnosis more accurately? Please add it like a previous study by Lin et al. (2015) to identify the enterovirus infection in the NHIRD.

Author Response

Point 1: The basic question is interesting and the data are consistent with most of the notion following revision. However, there are still some minor points need to be addressed.

Response 1: Thank you for the approval. We have revised the manucirpt based on the comments.

Point 2: Headline: " Epidemiologic implication of the association between herpes simplex virus infection and the risk of type 1 diabetes mellitus: a nationwide case-control study in Taiwan " - might be a better title in my personal opinion.

Response 2: Thank you for the comment. We modified the headline with the reviewer’s suggestion.

Point 3: Is there any reference about the criteria to support HHV diagnosis more accurately? Please add it like a previous study by Lin et al. (2015) to identify the enterovirus infection in the NHIRD.

Response 3: Thank you for the comment. The ICD-9-CM were regularly reviewed by the the Bureau of NHI. Diseases with the incorrect coding are not be reimbursed and the institutions will be fined. The high accuracy of data in the THIRD have been proved by previous studies. We have added references to support HHV diagnosis more accurately in the revised manuscript (Line 285-287) and show below:

…The diagnosis of HHV infection is based on the diagnosis code of ICD-9-CM. No laboratory data were available to support the diagnosis owing to the absence of such records in the database. Therefore, the accuracy of diagnosis may be questionable to some degree; however, the data provided from the NHIRD are high accurate which have been proved by previous studies [43, 44]. The Bureau of NHI endeavored the quality of NHIRD [45].

  1. Cheng, C.L.; Kao, Y.H.Y.; Lin, S.J.; Lee, C.H.; Lai, M.L. Validation of the National Health Insurance Research Database with ischemic stroke cases in Taiwan. Pharmacoepidemiol Drug Saf.,2011, 20(3), 236-242.
  2. Yu, Y. B., Gau, J. P., Liu, C. Y., Yang, M. H., Chiang, S. C., Hsu, H. C., ... & Chen, T. J. (2012). A nation-wide analysis of venous thromboembolism in 497,180 cancer patients with the development and validation of a risk-stratification scoring system. Thrombosis and haemostasis, 108(08), 225-235.
  3. Hsieh, C.Y.; Su, C.C.; Shao, S.C.; Sung, S.F.; Lin, S.J.; Yang, Y.H.K.; Lai, E.C.C. Taiwan’s national health insurance research database: past and future. Clin Epidemiol. 2019, 11, 349–358.
